# Qualitative process study to explore the perceived burdens and benefits of a digital intervention for self-managing high blood pressure in Primary Care in the UK

Katherine Morton,[1] Laura Dennison,[1] Katherine Bradbury,[1] Rebecca Jane Band,[1] Carl May,[2] James Raftery,[3] Paul Little,[4] Richard J McManus,[5] Lucy Yardley[1]

[1]Academic Unit of Psychology, University of Southampton, Southampton, UK
[2]Faculty of Health Sciences, University of Southampton, Southampton, UK
[3]Faculty of Medicine, Southampton University, Southampton, UK
[4]Primary Care Research, University of Southampton, Southampton, UK
[5]Nuffield Department of Primary Care Health Sciences, University of Oxford, Oxford, UK

**Correspondence to**
Katherine Morton;
ksm1r13@soton.ac.uk

## ABSTRACT

**Objectives** Digital interventions can change patients' experiences of managing their health, either creating additional burden or improving their experience of healthcare. This qualitative study aimed to explore perceived burdens and benefits for patients using a digital self-management intervention for reducing high blood pressure. A secondary aim was to further our understanding of how best to capture burdens and benefits when evaluating health interventions.

**Design** Inductive qualitative process study nested in a randomised controlled trial.

**Setting** Primary Care in the UK.

**Participants** 35 participants taking antihypertensive medication and with uncontrolled blood pressure at baseline participated in semistructured telephone interviews.

**Intervention** Digital self-management intervention to support blood pressure self-monitoring and medication change when recommended by the healthcare professional.

**Analysis** Data were analysed using inductive thematic analysis with techniques from grounded theory.

**Results** Seven themes were developed which reflected perceived burdens and benefits of using the intervention, including worry about health, uncertainty about self-monitoring and reassurance. The analysis showed how beliefs about their condition and treatment appeared to influence participants' appraisal of the value of the intervention. This suggested that considering illness and treatment perceptions in Burden of Treatment theory could further our understanding of how individuals appraise the personal costs and benefits of self-managing their health.

**Conclusions** Patients' appraisal of the burden or benefit of using a complex self-management intervention seemed to be influenced by experiences within the intervention (such as perceived availability of support) and beliefs about their condition and treatment (such as perceived control and risk of side effects). Developing our ability to adequately capture these salient burdens and benefits for patients could help enhance evaluation of self-management interventions in the future. Many participants perceived important benefits from using the intervention, highlighting the need for theory to recognise that engaging

## Strengths and limitations of this study

► The exploratory, open approach to data collection enabled us to capture whichever benefits or burdens were most salient to the participants.
► We only interviewed participants at one point in time, so we were unable to gain an understanding of dynamic changes in perceived benefits or burdens over time.
► Both well and poorly controlled hypertensive patients took part in the interviews, but it was difficult to recruit low users of the intervention which could limit the generalisability of the findings.
► The asymptomatic nature of hypertension and the unique medication change pathway means that these findings may not be generalisable across conditions.

in self-management can include positive as well as negative aspects.

**Trial registration number** ISRCTN13790648; Pre-results.

## BACKGROUND

The work involved in looking after one's health when living with a chronic condition can include complex tasks such as organising and adhering to treatment regimens, interacting with healthcare professionals (HCP), regular monitoring of health indicators and making health-related decisions, all of which can accumulate into a considerable burden.[1] Digital self-management interventions are often developed to improve health outcomes, but these interventions could also either increase or minimise the burden of the healthcare process for patients. Developing our understanding of the burdens of self-management can help to better optimise the delivery of healthcare to improve adherence and well-being.[1–3] Burden of Treatment

(BoT) theory provides a mechanism for understanding these experiences in the context of patients' personal capacity to cope, with emphasis on the role of wider healthcare systems and social networks available to the patient.[1]

Health economic evaluations also focus on understanding the impact of healthcare on patients, seeking to weigh up the resources used against the health outcomes in order to better inform decision-making. Recent guidelines for economic evaluations in health and medicine recommend adopting a societal perspective such that all relevant outcomes are evaluated, rather than focusing only on formal healthcare costs.[4] In particular, personal costs such as time spent in self-care should be included. Consequently, BoT theory and health economic evaluations share an interest in adequately capturing the wider burdens or personal costs of engaging with healthcare. For consistency in terminology in this paper, negative outcomes/personal costs of healthcare will be referred to as 'burdens'.

BoT theory considers patients' time as a resource that is used by the healthcare system, while health economic evaluation counts time as an 'opportunity cost' where the patient 'spends' time that could have been spent on something other than healthcare. However, subjective experiences of time spent on digital interventions may be varied and complex. Heterogeneity in the relative value placed on the outcomes of the intervention[5] may mean that for some participants the time spent engaging with elements of an intervention is not perceived as a burden but rather as a benefit, either because it is interesting, pleasant or meaningful in and of itself or because of the positive outcomes it can lead to. In other words, some people may actually like engaging with healthcare. The value of exploring the personal benefits of intervention participation has not received as much focus as understanding the costs, such as treatment burden. McNamee et al[6] proposed that the health research guidelines for economic analysis may need to be adjusted for digital health interventions to ensure we can fully capture the heterogeneous costs and benefits arising when complex interventions are implemented in complex systems.

To further our understanding of how patients perceive benefits and burdens when using digital health interventions, we carried out a qualitative process study.[7] The digital Home and Online Management and Evaluation of Blood Pressure (HOME BP) intervention was developed based on best practice recommendations to help improve hypertension in poorly controlled patients by facilitating self-monitoring of blood pressure (BP) at home and prompting appropriate intensification of medication by HCPs.[8] This intervention could help minimise the treatment burden of hypertension by providing an online healthcare system in which HCPs have sight of patients' home readings, streamlining the process for finding the most effective medication without the need for attending the general practitioner (GP) surgery. However, HOME BP is a complex, interactive multicomponent intervention, which creates potential diversity in the perceived burden and benefits for participants using it. The contexts in which the intervention is embedded may also be diverse, and factors such as individual differences in patients' health status, beliefs about medication and risks of high BP, availability of time and resources, and access to support may influence how the intervention is perceived and valued. The HOME BP intervention was developed using the person-based approach[9] which emphasises the importance of understanding participants' unique perspectives and different situations when developing and implementing digital interventions. Adopting a more granular approach to the evaluation of benefit and burden is consistent with the person-based approach, and with the BoT approach of fully understanding the participants' perspective.

The present study aimed to explore the perceived burden and benefits of using a digital health intervention for self-managing BP using qualitative process interviews with intervention and usual care participants taking part in a randomised controlled trial (RCT). This paper seeks to interpret the implications for optimising the capture of perceived costs and benefits in health economic evaluations and evaluating the burden of treatment.

## METHODS

### Design

A qualitative process study embedded in the HOME BP trial[8] was approved by the University of Southampton and NHS Research Ethics committees. The COREQ checklist (Consolidated criteria for reporting qualitative studies) was used to ensure comprehensive reporting of the study[10] (online supplementary file 1).

### Intervention

The HOME BP programme supported participants to self-manage their high BP, primarily via home self-monitoring of BP and making changes to dose/drug type when recommended by the HCP. Lifestyle change modules were also available, but optional as the key target behaviours for the intervention were self-monitoring and medication change adherence.[8 11] Participants using HOME BP were supported by a 'prescriber' (GP or nurse prescriber responsible for changing medication) and a 'supporter' (nurse or healthcare assistant who supported participants in self-monitoring and choosing lifestyle changes).

Participants were invited to use the online programme by their GP and were randomised to usual care or intervention after completing baseline measures online. Those randomised to the intervention group completed two online training sessions which sought to overcome concerns about variability in readings and changing medication. Participants were encouraged to monitor their BP in the mornings, but the programme allowed flexibility as it was most important that people found a time of day that suited them to monitor their BP. Both intervention

**Table 1** HOME BP intervention characteristics

| Target behaviour | Description |
|---|---|
| Self-monitoring BP | Participants monitored their BP at home for 7 days every 4 weeks. After 7 days, they entered their BP readings on the HOME BP website and received instant automated feedback using a traffic light system. If BP was very high (red) or very low (blue), they were told to contact their GP surgery. If BP was above target (amber), they were told their prescriber would contact them about a medication change. If BP was on target (green), they were congratulated and asked to monitor their BP again next time. |
| Medication change | The prescriber planned three potential medication changes with the participant at the start of the study. HOME BP informed prescribers by email when a patient's home BP readings were above target and they could implement a preplanned change without needing to see the participant for an appointment. |
| Optional lifestyle changes | At 9 weeks after randomisation, participants had the option of choosing an online session to support lifestyle change to help control their BP, specifically weight management, salt reduction, healthy diet, physical activity or alcohol reduction. Participants were alerted by email when this became available, and saw an option to view the healthy lifestyles session each time they logged on to HOME BP. The online lifestyle change sessions could be started at any time during the 12-month trial, from 9 weeks. |

BP, blood pressure; GP, general practitioner; HOME BP, Home and Online Management and Evaluation of Blood Pressure.

and usual care participants were followed up at 6 and 12 months after randomisation.

Table 1 describes the HOME BP intervention in more detail.

### Participants

Patients were eligible to take part in the HOME BP trial if they had uncontrolled hypertension managed in Primary Care (mean BP reading of 140/90 mm Hg or more at baseline taken at the GP surgery using a validated electronic automated sphygmomanometer (BP TRU BPM-200)). In addition, they needed to be prescribed one to three antihypertensive medications at baseline, and aged over 18 (full inclusion and exclusion criteria are listed in the protocol[8]).

Both intervention and usual care participants were invited to take part in interviews as we felt that obtaining an understanding of managing BP in usual care would aid interpretation of the perceived burden and benefits of the intervention. We aimed to speak to participants at a range of time-points during the 12-month trial from 10 weeks onwards as this gave participants the opportunity to become familiar with HOME BP. No new intervention content was introduced after the lifestyle sessions became available at 9 weeks.

### Recruitment and interview procedure

A subsample of RCT participants were invited by email to provide feedback on their experiences of managing their BP (n=78 of 622 patients in the RCT). Informed consent was taken by post or online, depending on participant preference. Recruitment was initially opportunistic, but subsequently a purposive approach was adopted to target younger participants, low engagers and those with recent uncontrolled self-monitored BP readings, informed by the concurrent analysis. Recruitment was stopped once the researchers agreed that data saturation had been reached and no new burdens or benefits were arising.

Semistructured interview schedules were codeveloped by experts in health psychology (KM, KB, RJB, LY, LD), health economics (JR) and sociology (CM). Open, inductive questions were carefully selected to elicit data about the burden and benefits of BP management perceived as most salient by the participants (see online supplementary file 2 for interview schedules). The interviews were conducted by telephone to minimise the burden on participants, except in one case where the participant asked to meet face-to-face due to struggling with hearing on the telephone. The interviews took place between February 2016 and February 2017. Each participant was given a £10 gift voucher to thank them for their time.

All interviews were conducted by KM (MSc, BSc termed 'the researcher'), a female PhD candidate in Health Psychology who was also employed as a research assistant. Each interview was audio-recorded, and the researcher also took notes and completed a self-reflection log afterwards to record any emerging thoughts on the data. Audio-recordings were transcribed verbatim and checked thoroughly by the researcher.

### Patient and public involvement

Patient and public involvement (PPI) representatives have been involved in the design and conduct of the RCT, including decisions about recruitment processes, outcome measures and trial procedures. We also discussed the findings of this qualitative process study with our PPI to facilitate our interpretations of the data. The participants in the study were patients, ensuring we were collecting experiences of burden from the target population, and the results were fed back to the study participants as a newsletter.

### Analysis

The analysis was an iterative process led by KM, supported by frequent discussion of emerging themes with LY and LD (who have extensive experience in qualitative research)

along with input regarding health economic and sociological perspectives (JR and CM). Inductive thematic analysis methods were used[12 13] with techniques from grounded theory such as memoing, constant comparison and diagramming to enhance our understanding and facilitate the development of higher themes.[14 15] Data collection and analysis ran concurrently to enable purposive sampling based on analytic insights. Thorough line-by-line coding was undertaken in NVivo V.10, 2017, and a coding manual was developed which evolved as more data were collected and coded. The emerging codes were constantly checked against the raw data to ensure the analysis was driven by the participants' own language and experiences.

All data relating to burdens and benefits of managing BP were analysed. We also coded factors that appeared to influence perceptions of burdens and benefits to facilitate an in-depth understanding of how participants appraised the intervention's value. A broad and open definition was adopted where benefits and burdens were defined as positive and negative outcomes or experiences of engaging in the intervention,[16] in order to facilitate a comprehensive representation of all potentially relevant data.

## RESULTS
### Participant characteristics
In the intervention group, 28 of 54 invited participants agreed to be interviewed (52%). In the usual care group, 7 of 24 invited participants agreed (29%). Most participants who did not take part chose not to reply, but those who did said they did not have anything to report on the trial (n=3 in usual care). The participants were from 19 different GP surgeries. Table 2 shows the sociodemographic and intervention details of the sample.

### Themes
Table 3 presents seven themes exploring perceived burdens and benefits of the HOME BP intervention. One metatheme also emerged concerning how illness and treatment beliefs about high BP appeared to influence participants' perceptions about the intervention's burdens and benefits, and this is discussed in relation to each theme it applies to. Figure 1 shows how illness and treatment perceptions about BP appeared to relate to the subthemes identified by the thematic analysis.

Where quotes are included, participants are referred to as 'p' followed by a number. Study group (intervention or usual care) is also included to help understand the quotes in context.

### Benefit of reassurance from seeing BP readings
*Reassurance when BP readings are well controlled*
Seeing well-controlled readings when self-monitoring BP gave participants peace of mind which was widely perceived as a benefit of the intervention. People described feeling relieved that their BP readings were lower than at the GP

**Table 2** Sociodemographic and intervention participant data (n=35)

| | Intervention participants | Usual care participants |
|---|---|---|
| n | 28 | 7 |
| Median duration of interview (range) | 38 (15–67) min | 28 (22–40) min |
| Median age (range) | 70 (41–87) years | 67 (52–77) years |
| Gender | 71% female | 43% female |
| Ethnicity | | |
| White | 24 | 6 |
| Black African | 1 | |
| Pakistani | 1 | |
| Other | 2 | 1 |
| Education levels | 9 No formal education | 2 No formal education |
| | 8 GCSE or A level | 3 GCSE or A level |
| | 10 Higher education | 1 Higher education |
| | 1 Other | 1 Other |
| Median number of weeks into the study at which the interview took place (range) | 20 (10–57) weeks | 17 (7–24) weeks |
| Poorly controlled BP at the time of the interview | 10/28 (36%) | NA* |
| Medication change recommended during the study | 15/28 (54%) | NA |
| Accessed optional healthy lifestyles session | 15/28 (54%) | NA |

*As BP self-monitoring was a key component of the intervention, BP readings were available for the intervention group throughout the duration of the study but data about BP from the usual care group were only available at RCT baseline and follow-up points. BP, blood pressure; GCSE, General Certificate of Secondary Education; NA, not applicable; RCT, randomised controlled trial.

surgery, and felt this gave them more insights into what their BP was like most of the time.

> What I do like about it is taking the blood pressure here at home, the readings are lower. And I find that quite reassuring that my blood pressure is not always high. (Intervention p11, well controlled)

Several usual care participants had decided to use their own BP monitors, and this group also described feeling reassurance when seeing their BP was well controlled.

### Reassurance from keeping an eye on BP
Most participants liked having an increased focus on their BP through regular monitoring and found it interesting

**Table 3** Themes and subthemes relating to perceived burdens and benefits of the intervention

| Themes | Subthemes | Exemplar participant quote |
|---|---|---|
| Benefit of reassurance from seeing BP readings | Reassurance when BP readings are well controlled | 'I'm so pleased. And my mind is at rest when we go on holidays and all that…I'm alright. I'm alright sort of thing. Yeah, peace of mind.' (Intervention p9, well controlled) |
| | Reassurance from keeping an eye on BP | 'It made me much more aware of what the problem is with the high blood pressure and by monitoring it so regularly, I know exactly where I stand with it.' (Intervention p15, well controlled) |
| Benefit of motivation for lifestyle change from seeing BP readings | Seeing BP readings motivated lifestyle change | 'It is quite interesting to see the effects of what I'm doing on the blood pressure and everything. So, I think that is—it is quite good.' (Intervention p18, well controlled) |
| Benefit of better health | Perceived health improvements from medication changes | 'It helped me to change my medication and then because of change of medication, my blood pressure went down. So definitely there is a benefit.' (Intervention p16, well controlled) |
| | Intervention can facilitate management of side effects | 'That medication didn't work, in that I was on holiday and my ankles swelled up so much—and my feet and my legs, so much so that I couldn't see my toes. So I stopped taking that medication. Was called back to the GP. And I'm now on a medication that works for me and is managing the blood pressure.' (Intervention p7, well controlled) |
| Burden of worrying about health | Negative emotional responses to seeing high readings | 'I was actually quite shocked because it was a—a lot higher.' (Intervention p6, poorly controlled) |
| | Worrying about medication change affecting health | 'I don't want to get more medication 'cause I'm already on a high dose and I don't want to increase it because it worries me about my kidneys.' (Intervention p24, poorly controlled) |
| Burden of uncertainty from self-monitoring | Uncertainty about whether readings are representative | 'If someone only ever takes it in the morning, and you tend to get those lower readings, are you really getting a true picture of what they're like in the afternoon or the evening?' (Intervention p10, well controlled) |
| | Uncertainty about what to do about high or low readings | 'I don't know what's going to happen in respect to that [amber feedback]. Whether I'm going to get a call from my GP, or whether he—so I'm a little bit, like, you know, in the air. I don't really know what's going to happen in that respect.' (Intervention p22, poorly controlled) |
| Burden of thinking about making healthy lifestyle changes | Worry or guilt about not engaging with healthy changes | 'I have looked at it [online healthy lifestyles session]. I wouldn't say I've looked at it seriously, and I need to.' (Intervention p4, poorly controlled) |
| Burden of the practicalities of adhering to intervention procedures | Burden of fitting self-monitoring into the day | 'I like to get up and have a cup of coffee and I'm thinking 'Well, let's get the blood pressure done first because otherwise I can't do that, you know, for a while afterwards.' So, I've found that quite—quite difficult.' (Intervention p5, poorly controlled) |

BP, blood pressure.

to compare their readings over time. However, one participant perceived that taking BP regularly could encourage too much attention on your health, which was a potential burden of the intervention for her (Intervention p28, BP control unknown as did not enter BP readings on HOME BP). This participant had low concern about her BP generally, and was not motivated to engage in self-management.

Even when participants had poorly controlled readings, many felt a benefit from the intervention as it enabled them to regularly check their BP and detect any problems instantly rather than carrying on unaware.

> I think it's helping me to know where my blood pressure stands because it's a regular thing every month. (Intervention p24, poorly controlled)

The knowledge that home readings were shared with the prescriber reassured participants as they knew that any problems would be detected and dealt with at the time, making them feel well cared for. This contrasted with the perceived burden of managing BP in usual care where some participants felt concerned that their GP did not change their medication when their home readings were too high, or would have liked more regular contact with their GP surgery to check their BP and medication.

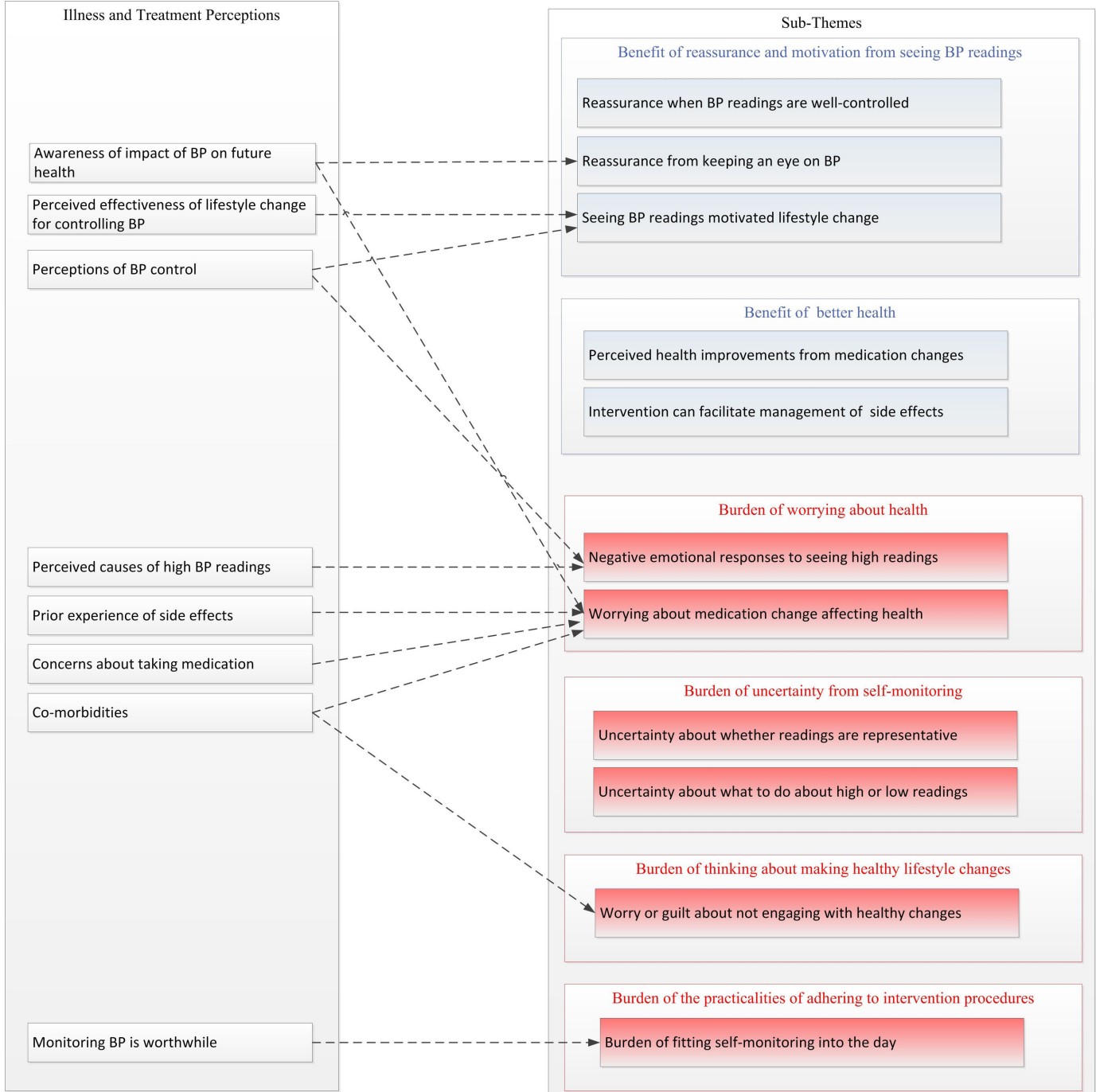

**Figure 1** Possible influences of illness and treatment beliefs on perceived burdens and benefits of the intervention. BP, blood pressure.

It would be nice to have it checked, I guess, you know, every three months or whatever. How—however often. I mean, how do they know that everything is working? (Usual care p4)

This shows that although participants in usual care gained reassurance from seeing low readings when they monitored at home, the lack of interaction with the GP surgery could cause concern when readings were high or when patients did not regularly monitor BP at home of their own accord.

**Benefit of motivation for lifestyle change from seeing BP readings**

Some participants were motivated to increase their physical activity, engage in stress management activities or healthy eating because they could see this had a positive impact on their BP readings. This helped them feel more in control of their BP.

By taking the readings regularly and frequently, it gave me more of a feedback straightaway if you like about anything, changes that I did make like a bit of exercise or…practicing relaxation and this sort of

thing. So that was quite nice, it was nice to feel that I was more in control of it again. (Intervention p20, well controlled)

Other participants felt frustrated after making lifestyle changes in the past which had no effect on their BP. This made them feel that lifestyle was ineffective for controlling BP.

I'm a completely different person. My diet's completely different. And my blood pressure remained the same. So I've done literally everything you physically possibly can to help yourself, and nothing's worked. (Intervention p1, well controlled)

### Benefit of better health
*Perceived health improvements from medication changes*
Many participants felt it was beneficial to change their medication when their readings were too high, and were very pleased when they perceived that a medication change led to lower BP readings because of the positive effect this would have on their health.

I've found that by having the medication changed up at regular intervals my blood pressure's improved all the time. (Intervention p15, well controlled)

A few participants felt that a medication change had not been effective at lowering their BP which could create doubt about their medication's effectiveness.

It's been doubled but it hasn't seemed to lower my blood pressure at all, in fact, it's at the same levels as it is sort of now, un-medicated. So I just think—I don't think it's the right one. You know, I can take the tablet but, actually, I don't think it's doing anything. (Intervention p26, poorly controlled)

### Intervention can facilitate management of side effects
Most participants did not experience any side effects from having their medication changed. Where side effects did occur, participants tended to perceive this as being a cost of taking medication (which was balanced against the benefit of controlling BP), rather than a burden of the intervention itself. They felt that the intervention could help them to be more aware of side effects, to identify alternative medications and to monitor how these affect their health.

That [side effect] would have happened, you know, no matter what. That would have been an issue but this has actually highlighted it, sort of, more clearly. (Intervention p5, poorly controlled)

### Burden of worrying about health
*Negative emotional responses to seeing high readings*
A burden of self-monitoring BP for some people was that seeing high readings could cause worry about health. Participants' beliefs about their BP control appeared to influence their appraisal of high readings. A few

participants believed their BP was well controlled, a belief which was perhaps reinforced by clinical staff approving their readings previously, and had only joined the study to help with research. These participants tended to feel shocked or annoyed when they received above-target feedback from the intervention as this challenged their beliefs.

At one time, I was told to go on medication, further medication, which I must admit I was not very happy about… When I used to go for a check with the nurse, if I'd have had those particular readings, they wouldn't have been high. (Intervention p17, poorly controlled)

Others were confused or frustrated by high BP readings when they could not understand why this might have happened.

I'm thinking about why my blood pressure has gone up. I can't think why. (Intervention p25, poorly controlled)

Meanwhile, people who expected to see high readings were less concerned because they had accepted that high readings were likely.

Just par for the course. It's what I expect from my blood pressure, really, so, it never worries me. (Intervention p5, poorly controlled)

Perceptions about the causes of high BP also influenced how anxious people felt about seeing high readings. Those who felt that high readings held serious implications for their health tended to feel frightened. Some even felt apprehensive *before* self-monitoring in case their readings were out of range, as they did not want to see evidence that their BP was too high or low.

Before I take my blood pressure, I do get stressed. I wouldn't say I get massively stressed because obviously I'm used to doing it now but … it's just that apprehension and thinking 'Oh, God, I hope it's not too high today.' I wonder really what's going on and how serious this is. (Intervention p26, poorly controlled)

Other people were able to dismiss one-off high readings without feeling anxious as they attributed high readings to less threatening explanations such as feeling stressed, not sitting still for long enough, positioning of the cuff or held a prior expectation of it being normal for BP to fluctuate. In these cases, the high readings had less negative emotional impact as they were not interpreted as indicating a serious underlying health issue.

### Worrying about medication change affecting health
Some participants were worried about the effects that changing BP medication could have on their health. Previous experience of side effects, existence of comorbidities and concerns about medication dependency or impact on kidneys tended to make participants feel more worried about changing medication.

Perceptions about the health risk of high BP in terms of stroke and cardiovascular disease tended to affect how burdensome participants perceived a medication change to be. Anxiety about future health could over-ride concerns about medication side effects or dependency as the behaviour was evaluated as beneficial in order to bring BP down, although sometimes participants still experienced conflict between the perceived benefit and burden.

> The blood pressure has gone down but now my worries have changed from blood pressure to other things. One is actually depending on medicine whole of my life. And secondly impact of medicine on my body like kidneys. (Intervention p16, well controlled)

### Burden of uncertainty from self-monitoring
*Uncertainty about whether readings are representative*

While some participants were confident making decisions about when to monitor their BP, others were worried about whether their readings were representative, especially when BP was seen to vary at different times of day or after physical activity or drinking coffee. This could lead to doubt about the meaningfulness of self-monitoring and the recommendations of the intervention.

> I wonder if maybe the time of day I'm doing it, maybe my blood pressure's always gonna be roughly that. And could it be different during the day, is the sort of thing that does play in my mind a bit. (Intervention p1, well controlled)

*Uncertainty about what to do about high or low readings*

Uncertainty could also become a burden after seeing an out-of-range BP reading, as the participant had to decide what to do next. This burden was removed when the prescriber provided quick, personalised feedback to the participant, but when they did not receive any contact from their prescriber or felt the prescriber was not available to provide support, this could create a feeling of doubt.

> I suppose I knew there was nothing to worry about but it's always a bit of a niggle in the back of your mind… even the days she's [the nurse prescriber] at work I can't ring her at work because she may be, you know, doing something else. (Intervention p21, well controlled)

### Burden of thinking about making healthy lifestyle changes
*Worry or guilt about not engaging with healthy changes*

Several participants felt they would like to lose weight, eat more healthily or do more physical activities but lacked the motivation or self-efficacy to make these changes, especially if they had other comorbidities. This could create feelings of guilt or worry about their failure to make healthy changes, which was a burden of the intervention for them.

> I understand that, obviously, I need to get my blood pressure down because it is very dangerously high, but I just don't know what to do about it, you know?… where I feel fatigued and worn out, I don't feel well enough at the moment to do any exercise. (Intervention p26, poorly controlled)

### Burden of the practicalities of adhering to intervention procedures
*Burden of fitting self-monitoring into the day*

Many participants felt that self-monitoring was easy to fit into their day, and some described this as being easier than going to the GP surgery to have their BP taken. Those with busy daily lifestyles tended to find it harder to remember to self-monitor, and a burden for some participants was deciding how best to fit self-monitoring into their routine given the instructions about not drinking coffee or exercising beforehand.

The perceived burden of regular self-monitoring seemed to be mitigated by the perceived benefit of the behaviour, such that those who felt reassurance from seeing low readings or with high motivation to control BP found it less hassle and easier to remember than those who felt anxious about self-monitoring or had only joined the study to help with research.

> There was no big deal. It doesn't take long and it's— it's quite nice to sit down and have a relax during the day. (Intervention p8, well controlled)

## DISCUSSION

This qualitative study has identified diverse perceived burdens and benefits of using a self-management digital intervention for high BP. In support of the BoT theory,[1] the HOME BP intervention appeared to reduce the burden on patients to self-manage their condition by improving access to regular HCP support and facilitating better understanding of their condition, but in some cases there was a burden of worry about health or changing medication. How much benefit a patient perceived from the intervention compared with burden seemed to be influenced by the dynamics of the patient–HCP interaction (described as 'Improving Cooperation' in BoT theory) and the patient's own resources to manage their condition and cope with medication (described as 'Capacity').

Another important factor relating to the burden experienced was personal beliefs about BP and treatment. Those who recognised that their BP was too high and did not have concerns about side effects or taking medication appeared to have more positive experiences of the intervention, perceiving self-monitoring as more worthwhile and feeling less anxious about seeing high readings or changing medication. This is consistent with the necessity-concerns framework.[17] BoT theory states

that people who are better equipped with resources and are more resilient may cope better with the burden imposed by healthcare,[18] but the importance of an individual's personal conceptualisation of their condition in how burdensome they find self-care is not strongly represented. This beliefs system may be partly encompassed by the 'Relational Integration' aspect of BoT theory, which refers to the extent to which patients trust the tasks they do for healthcare (eg, self-monitoring BP), and feel confident in the outcomes of these tasks (eg, changing medication). However, illness and treatment perceptions[19] are not explicitly covered by the theory and it may be helpful to consider them as additional factors which might influence the experience of treatment burden.

### Implications for measurement of benefit and burden

The present study demonstrates the value of collecting in-depth qualitative data to develop a detailed understanding of the burden of treatment, and to discover perceptions specific to the context in which the intervention was implemented. The important psychosocial outcomes discovered using qualitative research can inform the selection or development of relevant quantitative measures to capture these factors in further evaluation.

Quantitative measures have been developed to appraise the structural aspects of burden of treatment,[20 21] but these are not intended to assess psychosocial factors such as reassurance, anxiety or uncertainty which this study suggests can influence the extent to which using an intervention is experienced subjectively as a burden.

Future research could explore how best to capture the perceived burden or benefit of an intervention. One approach might be to simply ask participants to quantify the net subjective burden or benefit of interventions. However, it could be challenging for participants to weigh complex heterogeneous psychosocial outcomes against one another and decide overall whether an intervention was more burdensome or beneficial. Capturing the extent to which patients experience positive or negative psychosocial outcomes might better assess how beneficial or burdensome the intervention was perceived to be. Although this would not produce a single outcome measure, cost-consequence analysis can be used to inform decision-making when an intervention has multiple relevant outcomes which cannot be aggregated into one value.[22] Coast et al[23] discuss whether a multidimensional approach is more informative for economic analysis or if a single aggregated value is more pragmatic.

Extending the evaluation of outcomes beyond health is in line with the capability approach,[24] which focuses on broader aspects of subjective well-being which are not assessed by generic measures such as the EQ-5D (five items for assessing standardised health status).[25] Tools used to capture perceived capability (such as the ICECAP,

a capability measure for use in economic evaluation[26] and ASCOT, a tool for evaluating social-care related quality of life)[27] are gaining support as holistic measures of economic evaluation, but do not assess the more specific psychosocial burdens and benefits of healthcare raised by participants in this study. Process utility emphasises the need to quantitatively measure the value that people attach to healthcare delivery. This approach might be relevant for evaluating how much value people perceive in the process of using digital health interventions and the capability this achieves.[28] It has been argued that process utility measures should also ask about the reasons behind patients' valuations, to better inform the decision-maker.[29] This would help capture the individual differences found in this study in how people appraise the personal value of a digital intervention, informed by their underlying illness and treatment beliefs.

### Strengths and limitations

A strength of the study was that we used relatively open questions formulated by a multidisciplinary team which enabled us to elicit and explore a wide range of perceived burdens and benefits, some of which were not anticipated at the outset of research. We are aware of the lead researcher's potential influence on the data analysis, which we strived to minimise by transparent memoing of decisions and regular team meetings to discuss the emerging themes. Participants were sent newsletters to describe the findings of the study, but were not invited to provide feedback on the analysis.

We succeeded in speaking to well-controlled and poorly controlled hypertensive participants at different points in the intervention, and there was a wide range of demographics in terms of age, education level and gender in the sample. However, the uptake rate from those invited to interviews was not high, particularly in the usual care group. Perhaps unsurprisingly, it was difficult to recruit low engagers in the intervention group, which could have helped reach theoretical saturation. In terms of wider applicability, we are aware that these findings may not be generalisable across other health conditions, as the lack of symptoms in hypertension and the stepped pathway for changing medication are quite unique features of this condition.

Repeated interviews with the same participants may have offered more insights into the dynamic nature of perceived burdens and benefits over time, although more regular conversations about the target behaviour could have influenced participants' BP management behaviour, therefore threatening the RCT conclusions. It has been noted that a key issue with process evaluations of interventions is the tendency for intervention content and impact to change over time,[7] such that deciding the optimal point to collect evaluation data is challenging.

Some of the burdens and benefits described by patients in this study were also found to a lesser extent in the qualitative development of the HOME BP

intervention, such as reassurance from seeing well-controlled readings, and some concerns about side effects and high or variable readings (Bradbury *et al*, Submission, 2017). Others were novel and only arose when participants experienced the full HOME BP intervention during the RCT as opposed to a prototype, for example, the perceived health improvements from medication changes. This demonstrates the value of conducting inductive qualitative research to explore users' perspectives at each stage of intervention development and evaluation, in line with the person-based approach.[9]

## CONCLUSIONS

In the context of this digital intervention, the study shows that participants' appraisal of burdens and benefits appeared to be influenced by both intervention factors, such as BP readings and perceived availability of the HCP, and patient characteristics, such as perceptions of BP control, previous experience of side effects and comorbidities. This nuanced evaluation would be lost in a population-level analysis, demonstrating the advantage of a more individualised approach for better understanding participants' perspectives of an intervention and how best to minimise the burden of treatment.

The study develops the recommendations of McNamee *et al*[6] that complex digital health interventions warrant a wider perspective for measuring health outcomes, and discusses the implications of capturing broader psychosocial outcomes for BoT theory and health economic evaluations.

The finding that some participants perceived personal benefits from using the intervention demonstrates that the process of healthcare can, in itself, be positive for some people, highlighting the importance of capturing transient short-term benefits to take these into account as well as the burden of self-management.

**Acknowledgements** We thank all the participants who took part in this research, and the PPI who helped in the design and conduct of the research.

**Contributors** KM recruited participants, conducted interviews, analysed data and wrote the manuscript. LD and KB contributed to study design and data analysis. KB and RJB developed the digital intervention. PL and RJM contributed to intervention development and interpretation of themes. CM and JR contributed to theoretical and methodological implications of the study findings. LY contributed to study design, data collection, data analysis and interpretation. All authors contributed to the manuscript preparation and provided final approval of the version to be published.

**Funding** This independent research was funded by the National Institute for Health Research (NIHR) Programme Grants for Applied Research Programme (grant reference number RP-PG-1211-20001). RJM is funded by an NIHR Professorship (NIHR-RP-R2-12-015) and the NIHR Oxford CLAHRC.

**Disclaimer** The views expressed are those of the author(s) and not necessarily those of the NHS, the NIHR or the Department of Health.

**Competing interests** RJM has received BP monitors for research purposes from Omron and Lloyds Pharmacies.

**Patient consent** Detail has been removed from this case description/these case descriptions to ensure anonymity. The editors and reviewers have seen the detailed information available and are satisfied that the information backs up the case the authors are making.

**Ethics approval** This qualitative process study was embedded in the HOME BP trial and approved by the University of Southampton and NHS Hampshire A Research Ethics committees (REC Reference 15/SC/0082).

**Provenance and peer review** Not commissioned; externally peer reviewed.

**Data sharing statement** Requests for data sharing can be sent to the corresponding author. Full transcripts of interviews are not available to protect participants' anonymity.

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
