## [Reviewer comments · BMJ Open]

ARTICLE DETAILS

TITLE (PROVISIONAL)	A qualitative process study to explore the perceived burdens and benefits of a digital intervention for self-managing high blood pressure in Primary Care in the UK.
AUTHORS	Morton, Katherine; Dennison, Laura; Bradbury, Katherine; Band, Rebecca; May, Carl; Raftery, James; Little, Paul; McManus, Richard; Yardley, Lucy

VERSION 1 – REVIEW

REVIEWER	Karin Kjellgren Department of Medical and Health Sciences, Linköping University, Sweden
REVIEW RETURNED	11-Dec-2017

GENERAL COMMENTS	The paper reports on perceived burdens and benefits for patients using an online self-management intervention for high blood pressure. This is a timely and important research area for improvement of quality in treatment of hypertensive patients. The paper is well written and relies on a well-structured research protocol. However, I have a few comments that need clarification from the author(s) to further improve the paper. Comments:  Title: "Perceived burdens and benefits of self-management interventions: A qualitative process study of an online intervention for self-managing high blood pressure" The paper reports from one study. Therefore, the first part of the title should be deleted. There are no results from self-management interventions. (plural) Abstract: Please clarify:  Line 23, be more specific about online self-management Line 24 ... how to best capture these outcomes, which outcomes? Line 36 A model was developed ... A model is not reported as a result in this paper. Are you referring to a theory or a model, the BoT? Strengths and limitations:  The sentence "Qualitative data is not commonly
---

	used in health economics evaluation, so further work would be needed to understand how relevant outcomes could best be captured quantitatively” on line 61 does not provide any additional information from the paper and could be removed.  4. Some sentences in the paper tend to be very long and would benefit from being rephrased, for example line 35-39, line 450-454. 5. Line 100 Provide references related to the method for the qualitative process study. Would ref. 29 be appropriate here? 6. I would like to see the authors clarify their use of the term “person-based” in relation to a “person-centered” approach. 7. Line 133 why was life-style changes optional? See recent guidelines for hypertension 2017. 8. Table 1.  • Medication change. Is it correct that “prescribers are informed”? I thought that the prescribers conducted the prescriptions. • “Nine weeks after randomisation, participants have the option of choosing an online session to support lifestyle change” but line 149 reads “No new intervention content was introduced after nine weeks”. Please clarify! 9. Page 12, Table 3 and forward, when referring to quotes describe the meaning of p (Intervention p9), is p a participant or patient? 10. How was baseline BP measured? 11. Was the semi-structured interview schedule piloted? The schedule includes an array of questions, was it possible to get answers to all these questions in the time frame used for the interviews? 12. How were “the usual care participants” informed about the intervention? 13. What does 17 (7 to 24) weeks into the study mean for “the usual care participants”? 14. Line 332, ... concerning how patients were worried that the BP values were representative How were the participants informed about the online system? 15. Line 393, concerning outcomes see comment 2 line 24, do these outcomes only pertain to changing medication? 16. Line 408, “Future research could explore how best to capture this”. Please clarify this. 17. Line 436, did the newsletters affect the result of the study? 18. Line 461 I would like to see the authors clarify the terms; “online intervention” in relation to “digital intervention”, are these equivalent to one another? If not, clarify. If they are, I would advise that only one term is used consistently throughout the paper.
--	---

REVIEWER	Dr Leanne Chalmers Curtin University, Australia
REVIEW RETURNED	05-Jan-2018

GENERAL COMMENTS	Many thanks to the authors for presenting this well-written and interesting manuscript. While many researchers investigating self-monitoring interventions, and in fact many clinicians, would have encountered some of the beliefs and attitudes revealed in this work,
--

	it was useful to see them mapped against the Burden of Treatment framework and the authors make some interesting suggestions regarding the implications of their work for evaluation of health outcomes. This manuscript was a pleasure to read. I have only a few minor suggestions: 1. The manuscript describes a qualitative study of a single digital self-monitoring intervention relating to a typically asymptomatic condition. Furthermore, the medications available to treat hypertension are typically well-tolerated, and if patients experience intolerable side effects, there are usually alternatives available. This is not typical of the majority of chronic health conditions. I would therefore suggest that the authors review the first part of the title of the paper (“Perceived burdens and benefits of self-management interventions” suggests a generalisability outside the current intervention) and also consider addressing this issue as a limitation in the Discussion. 2. Abstract  - The aims should typically be in past tense. - Line 44: ‘in the future’ - Line 46: ‘for theory to allow that engaging in self-management’: this wording is confusing for the naïve reader; please review - Line 61: ‘Qualitative data are...’ 3. Methods  - Table 1: this may read better in past tense. - p8, line 143: ‘see7’ – please consider rewording. - p8, line 151: Please consider indicating the total number of control and intervention participants involved in the trial to provide context about its overall size and scope. 4. Results  - p10, line 191: It is not usual to commence a sentence with a number; please review. 5. General  - Please be consistent with the use of abbreviations – e.g. HCPs is defined in p4 (line 68) but then used in full on p5 (line 103); also Home BP instead of HOME BP (p6, line 125). - Please also review the use of the term ‘GP Practice’ – this reads as ‘general practice practice’. - p24, line 387 and p25, line 402: please review typos. - Please review consistency of abbreviation of journal titles in the References.
--	--

VERSION 1 – AUTHOR RESPONSE

EDITORIAL REQUIREMENTS

1. Please revise your title to state the research question, study design, and setting (location). This is the preferred format for the journal.

Title has been amended to: A qualitative process study to explore the perceived burdens and benefits of a digital intervention for self-managing high blood pressure in Primary Care.

2. Please revise the Strengths and Limitations section (after the abstract) to focus on the methodological strengths and limitations of your study rather than summarizing the results.

Two bullet points were removed which were descriptions of the findings. A new limitation has been added which relates to the generalisability of the findings.

REVIEWER 1 COMMENTS AND AUTHORS' RESPONSES

3. Title: "Perceived burdens and benefits of self-management interventions: A qualitative process study of an online intervention for self-managing high blood pressure". The paper reports from one study. Therefore, the first part of the title should be deleted.

There are no results from self-management interventions. (plural)

Agreed, the title has been amended to:

A qualitative process study to explore the perceived burdens and benefits of a digital intervention for self-managing high blood pressure in Primary Care.

4. Abstract:

Please clarify:

Line 23, be more specific about online self-management

The abstract now refers to a digital self-management intervention, rather online, for consistency of terminology.

Line 24 ... how to best capture these outcomes, which outcomes?

'these outcomes' has now been replaced with 'burdens and benefits' to be more specific.

Line 36 A model was developed ... A model is not reported as a result in this paper.

Are you referring to a theory or a model, the BoT?

Agreed. The Abstract now states that 'the analysis showed' (Line 36) rather than mentioning a model, which could be misleading.

5. Strengths and limitations:

The sentence "Qualitative data is not commonly used in health economics evaluation, so further work would be needed to understand how relevant outcomes could best be captured quantitatively" on line 61 does not provide any additional information from the paper and could be removed.

Agreed, this strength was removed.

6. Some sentences in the paper tend to be very long and would benefit from being rephrased, for example line 35-39, line 450-454

These sentences have been broken down into smaller sentences to improve ease of reading (now lines 35-37, and 463-465).

7. Line 100 Provide references related to the method for the qualitative process study. Would ref. 29 be appropriate here?

This reference has been added. (Line 97)

8. I would like to see the authors clarify their use of the term "person-based" in relation to a "person-centered" approach.

In the interest of manuscript length, the difference between person-based and person-centred approaches is not explored in the manuscript but the reference to the person-based approach paper

provides more detail about this. The person-based approach has grown out of a number of related and complimentary approaches, including person-centred therapy and user-centred design, but the authors felt it is not within the scope of this paper to discuss this. Readers who wish to know more can read the referenced paper which discusses the full wider context of the PBA.

9. Line 133 why was life-style changes optional? See recent guidelines for hypertension 2017. Lifestyle changes were optional because the key target behaviours for the intervention were self-monitoring and medication titration. This is now stated in the paper, with reference to the RCT protocol and planning paper where this decision is documented and referenced (line 130-131). The lifestyle changes were introduced later so as not to confront the participant with multiple behaviour changes at once, and to allow the self-monitoring behaviour to become more habitual before further changes were introduced.

10. Table 1.

Medication change. Is it correct that “prescribers are informed”? I thought that the prescribers conducted the prescriptions.

It is correct that prescribers were informed when a medication change was recommended. HOME BP alerted prescribers by email when patients’ home readings exceeded a threshold. They were prompted to issue the prescription for the medication change. The wording in Table 1 has been changed slightly to clarify this.

“Nine weeks after randomisation, participants have the option of choosing an online session to support lifestyle change” but line 149 reads “No new intervention content was introduced after nine weeks”. Please clarify!

The lifestyle change sessions became available exactly 9 weeks after randomisation. After this point, no new intervention content was introduced. Lines 155-156 have been re-worded slightly to clarify this point.

11. Page 12, Table 3 and forward, when referring to quotes describe the meaning of p (Intervention p9), is p a participant or patient? Reviewer 1

P stands for participant. This has been clarified in Line 216.

12. How was baseline BP measured?

Baseline and follow-up BP were taken at the GP Surgery using a validated electronic automated sphygmomanometer (BP TRU BPM 200). This has been added at Line 148.

13. Was the semi-structured interview schedule piloted? The schedule includes an array of questions, was it possible to get answers to all these questions in the time frame used for the interviews?

The semi-structured interview schedule was not piloted.

Yes it was possible to get answers to all questions.

14. How were “the usual care participants” informed about the intervention?

The usual care participants were fully informed of the intervention in the Participant Information Sheet. They were randomised online and the meaning of being in usual care was explained at the time of randomisation and in an email and postal letter.

This has been clarified on line 136.

15. What does 17 (7 to 24) weeks into the study mean for “the usual care participants”?

This is how many weeks post-randomisation they were interviewed. Usual care participants were followed up at 6 and 12-months.

This has been made clear in the manuscript by editing the variable name in the Table to 'number of weeks since randomisation', and clarifying in the Intervention section of the methods that both groups were followed-up at 6 and 12-months post-randomisation (line 141).

16. Line 332, ... concerning how patients were worried that the BP values were representative How were the participants informed about the online system?

Participants were invited to use the online programme by their GP, and completed two online training sessions at the start which sought to overcome concerns about variability in readings, medication change etc. Participants were encouraged to monitor their BP in the mornings, but the programme allowed flexibility as it was most important that people found a time of day that suited them to fit in BP monitoring.

This is now clarified in the Intervention section of the method (lines 137-140) to help contextualise the findings about uncertainty re representativeness of readings.

17. Line 393, concerning outcomes see comment 2 line 24, do these outcomes only pertain to changing medication?

No the study was interested in any burden or benefit introduced by self-monitoring BP, receiving automated feedback on readings, changing medication if required, entering their readings digitally etc. The participants who recognised that their BP was too high and were not concerned about side effects seemed to feel more positive about several elements of the intervention, e.g. perceiving self-monitoring as worthwhile, and feeling less anxious about seeing high readings, so it was not only about their perceptions of changing medication.

This is described in the manuscript on line 395 so no changes have been made.

18. Line 408, "Future research could explore how best to capture this". Please clarify this.

This sentence has been amended for clarity: "Future research could explore how best to capture the perceived burden or benefit of an intervention" (Now line 417).

19. Line 436, did the newsletters affect the result of the study?

No, the newsletters were sent out after a participant had taken part in an interview. Feedback from patients on the newsletters was not collected.

We have not changed the manuscript.

20. Line 461 I would like to see the authors clarify the terms; "online intervention" in relation to "digital intervention", are these equivalent to one another? If not, clarify. If they are, I would advise that only one term is used consistently throughout the paper

Agreed that this is confusing to switch between terms. The term 'digital intervention' is now used throughout, and it is specified that the readings are entered on a website in Table 1 which describes the intervention.

REVIEWER 2 COMMENTS AND AUTHORS' RESPONSES

21. The manuscript describes a qualitative study of a single digital self-monitoring intervention relating to a typically asymptomatic condition. Furthermore, the medications available to treat hypertension are typically well-tolerated, and if patients experience intolerable side effects, there are usually alternatives available. This is not typical of the majority of chronic health conditions.

I would therefore suggest that the authors review the first part of the title of the paper ("Perceived burdens and benefits of self-management interventions" suggests a generalisability outside the current intervention) and also consider addressing this issue as a limitation in the Discussion.

Agreed. The Title has been amended to:

A qualitative process study to explore the perceived burdens and benefits of a digital intervention for self-managing high blood pressure in Primary Care.

A limitation has been added about the generalisability of the findings to other conditions.

22. Abstract

- The aims should typically be in past tense.

- The aims have been changed to the past tense.

- Line 44: 'in the future'

- We have added 'the' to this line in the abstract.

- Line 46: 'for theory to allow that engaging in self-management': this wording is confusing for the naïve reader; please review

- We have changed this sentence to 'for theory to recognise that engaging in self-management...'.-

- Line 61: 'Qualitative data are...'

- We have changed to qualitative data are, although this strength was subsequently removed (see Reviewer 1 comments)

23. Methods

- Table 1: this may read better in past tense.

- Table 1: Agreed, this has been changed to past tense.

- p8, line 143: 'see7' – please consider rewording.

- Agreed, this now says "full inclusion and exclusion criteria are listed in the protocol (ref)"

- p8, line 151: Please consider indicating the total number of control and intervention participants involved in the trial to provide context about its overall size and scope.

- Agreed. Total number of RCT participants has been added on line 168.

24. Results

- p10, line 191: It is not usual to commence a sentence with a number; please review.

Agreed, this has been amended to:

In the intervention group, 28 of 54 invited participants agreed to be interviewed (52%). (line 198)

25. General

- Please be consistent with the use of abbreviations – e.g. HCPs is defined in p4 (line 68) but then used in full on p5 (line 103); also Home BP instead of HOME BP (p6, line 125).

Thank you for noticing inconsistent use of abbreviations.

- Healthcare professionals is only defined in full once at the start
- HOME BP is capitalised throughout.

- Please also review the use of the term 'GP Practice' – this reads as 'general practice practice'.

Agreed, 'GP Practice' has been amended to 'GP Surgery'.

- p24, line 387 and p25, line 402: please review typos.

The additional full stop was deleted on line 396 (previously line 387)

A space was added between 'research' and 'can' on line 411 (previously line 402).

- Please review consistency of abbreviation of journal titles in the References.

The references have been updated to use abbreviated journal titles consistently.

VERSION 2 – REVIEW

REVIEWER	Karin Kjellgren Department of Medical and Health Sciences, Linköping University, Linköping Sweden
REVIEW RETURNED	03-Feb-2018

GENERAL COMMENTS	The authors have answered the questions from reviewer 1 in a meritorious way
--

REVIEWER	Dr Leanne Chalmers Curtin University, Australia
REVIEW RETURNED	08-Feb-2018

GENERAL COMMENTS	Thank you to the authors for their careful and thoughtful responses to the reviewers' comments. I am satisfied that the more significant issues have been adequately addressed, and just have a very few minor suggestions, largely in the interests of readability: Background 1. Line 65: please replace 'regimes' with 'regimens'. Methods 2. Please change all of the 'Intervention' section (line 128 onwards) into past tense for consistency with the remainder of the paper. 3. Line 141: no need for dash in 'followed-up'. 4. Please be consistent with the capitalisation of 'GP surgery' (e.g. in Table 1) and 'GP Surgery' (e.g. lines 147 and 371). I would suggest that you go with the former. Similarly, I don't think that 'practice' is a proper noun (Results – lines 248 and 253). 5. Lines 146-150: this has become quite a long, wordy sentence. Please consider breaking it into two sentences, or at least a comma after '(BP TRU BPM 200))'. Results 6. Table 2: Would the patient characteristics be better summarised by median and range (rather than average and range), given the relatively small sample sizes? Please include what is being reported for 'Number of weeks since randomisation'. Discussion 7. The spacing of the paragraph commencing on line 448 appears to change. Please change 'generalisable' to British spelling in this paragraph (line 453).
---

VERSION 2 – AUTHOR RESPONSE

1. Please revise your title to include the location. This is the preferred format for the journal.(Editorial requirements)

Title has been amended to: A qualitative process study to explore the perceived burdens and benefits of a digital intervention for self-managing high blood pressure in Primary Care in the UK.

2. Line 65: please replace 'regimes' with 'regimens'. (Reviewer 2)

Replaced 'regimes' with 'regimens'

3. Please change all of the 'Intervention' section (line 128 onwards) into past tense for consistency with the remainder of the paper (Reviewer 2)

Intervention section is now written in past tense for consistency

4. Line 141: no need for dash in 'followed-up'.(Reviewer 2)

Removed the dash in 'followed-up'

5. Please be consistent with the capitalisation of 'GP surgery' (e.g. in Table 1) and 'GP Surgery' (e.g. lines 147 and 371). I would suggest that you go with the former. Similarly, I don't think that 'practice' is a proper noun (Results – lines 248 and 253). (Reviewer 2) GP surgery is now consistently capitalised throughout, with a lower case 's'

'Practice' has been changed to 'practice' on lines 248 and 253

6. Lines 146-150: this has become quite a long, wordy sentence. Please consider breaking it into two sentences, or at least a comma after '(BP TRU BPM 200)'.(Reviewer 2)

Agreed that this sentence had become quite hard to process. This has been broken down into 2 sentences and the order slightly changed to improve readability.

7. Table 2: Would the patient characteristics be better summarised by median and range (rather than average and range), given the relatively small sample sizes? Please include what is being reported for 'Number of weeks since randomisation'.(Reviewer 2)

Agreed that age, duration of interview and weeks into the study would be better summarised as medians and range. This change has been made to Table 2 for both groups, although the medians for the usual care group were the same as the means so there is no track changes to show this change. We have also explained more clearly what is being reported for 'number of weeks since randomisation'

8. The spacing of the paragraph commencing on line 448 appears to change. Please change 'generalisable' to British spelling in this paragraph (line 453).(Reviewer 2)

The spacing for this paragraph has been corrected, and 'generalisable' now has the British spelling.